# Development of Antibacterial Cotton Textiles by Deposition of Fe_2_O_3_ Nanoparticles Using Low-Temperature Plasma Sputtering

**DOI:** 10.3390/nano13243106

**Published:** 2023-12-09

**Authors:** Agnė Giedraitienė, Modestas Ružauskas, Rita Šiugždinienė, Simona Tučkutė, Kastytis Grigonis, Darius Milčius

**Affiliations:** 1Institute of Microbiology and Virology, Faculty of Veterinary Medicine, Lithuanian University of Health Sciences, LT-44307 Kaunas, Lithuania; modestas.ruzauskas@lsmuni.lt (M.R.); rita.siugzdiniene@lsmuni.lt (R.Š.); 2Center for Hydrogen Energy Technologies, Lithuanian Energy Institute, LT-44403 Kaunas, Lithuania; simona.tuckute@lei.lt; 3Visdenta Ltd., LT-50185 Kaunas, Lithuania; kastytis@visdenta.lt

**Keywords:** iron oxide, nanoparticles, cotton, antimicrobial activity

## Abstract

Antibacterial textiles can help prevent infections from antimicrobial-resistant pathogens without using antibiotics. This work aimed to enhance the cotton fabric’s antimicrobial properties by depositing Fe_2_O_3_ nanoparticles on both sides of its surface. The nanoparticles were deposited using low-temperature plasma technology in a pure oxygen atmosphere, which is environmentally friendly. The Fe_2_O_3_ nanoparticles formed clusters on the fabric surface, rather than thin films that could reduce the airflow of the textile. The optimal conditions for the nanoparticle deposition were 200 W of plasma power, 120 min of immersion time, and 5 cm of Fe cathode–textile sample distance. The received antimicrobial textile was tested and the high efficiency of developed materials were successfully demonstrated against 16 microbial strains (Gram-positive and Gram-negative bacteria and fungi).

## 1. Introduction

Microbe-resistant clothing is essential for environments that are sensitive to microbes, such as hospitals, laboratories, and pharmaceutical factories [1]. Cotton is a widely used fabric in the textile industry, as it has many desirable properties, such as durability, absorbency, color retention, softness, breathability, and comfort [2,3]. However, cotton lacks antimicrobial activity, which makes it vulnerable to microbial growth (bacteria, fungi), including odor-causing bacteria [4], and a potential source of viral and bacterial transmission [5]. To make cotton antiviral and antibacterial, functional additives are applied to its surface; cotton is an ideal material for integrating and incorporating nanomaterials [2]. Metal inorganic nanoparticles (MNPs) such as TiO_2_, SiO_2_, Cu_2_O, CuO, and AgO are among the most common and popular nanomaterials for nanotextile production, as they are resistant to high temperatures, stable under ultraviolet rays, and have low toxicity to humans [6]. The effectiveness of nanotextiles depends on the type and size of the metal nanoparticles, the deposition time, and other factors. Our previous research showed that the antimicrobial activity of a middle mask layer coated with CuO nanoparticles depended on the deposition time of CuO nanoparticles (at least 30 min). The coated mask layer had a bactericidal effect against both Gram-positive and Gram-negative bacteria but had little or no effect on yeast *Candida* spp. [7].

Scientists and industries are looking for new and less explored metal nanoparticles to modify textile surfaces. Most of the previous studies on nanotextiles used a single-sided coating of fabrics with nanoparticles [8]. However, there is limited information on fabrics coated with metal oxide nanoparticles on both sides. This technique can greatly enhance the antimicrobial activity of cotton.

Fe_2_O_3_ nanoparticles are a kind of iron oxide nanomaterial that have potential medical applications as antimicrobial agents [9]. Their antibacterial activity depends on various factors, such as their size, shape, surface charge, and coating methods. They can kill microbes by different mechanisms, such as producing reactive oxygen species, damaging bacterial cell membranes, and interfering with bacterial metabolism [10,11,12]. Fe_2_O_3_ NPs have shown good antibacterial activity against Gram-positive bacteria (*S. aureus*), Gram-negative bacteria (*E. coli* and *Vibrio fischeri*), fungi, and yeast [13]. Therefore, these nanoparticles could be useful for antimicrobial food packaging materials and medical devices [14]. Moreover, iron oxide nanoparticle-based systems could also be involved in drug delivery systems [15].

There are various methods and strategies for producing NPs, such as thermal evaporation, pulsed laser deposition, sputter deposition, CVD, and chemical/solution processes like coprecipitation, sol-gel, and solvothermal/hydrothermal methods [16]. Among them, magnetron sputtering uses plasma to sputter Fe_2_O_3_ targets and deposit the nanoparticles on nanofibers and textile substrates. This method can produce oxide nanoclusters and uniform coatings with excellent adhesion [17]. Surface functionalization, morphology, and thermal properties of polyamide6/O-MMT composite nanofibers can be improved by Fe_2_O_3_ sputter coating [17]. Another method for producing Fe_2_O_3_ nanomaterials with high precision is molecular beam epitaxy. However, this method has very high technological costs [18]. Fe_2_O_3_ nanoparticles can also be produced using wet chemical methods and combined with other materials, such as Ag, to enhance their antibacterial and antifungal activities [19].

This study used a cotton fabric treated with Fe_2_O_3_ nanoparticles on both sides, using low-temperature plasma deposition in an oxygen atmosphere with a 200 W power source and 120 min synthesis time. A low-temperature plasma approach does not require high vacuum conditions. The method is simple to use, relatively inexpensive compared to other PVD methods, and can be considered as a green chemistry method, as it only requires a pure Fe target, oxygen as working gas, and electricity to run the deposition processes. No residual waste is produced during the deposition of the nanoparticles on the textile surfaces.

In addition to the benefits, the textile surface is bombarded by ions from the plasma, which can help to increase the surface energy and the adherence of Fe_2_O_3_ nanoparticles to the textile surface [20]. We tested various technological conditions, starting from very low plasma power of 20 W and short immersion times of a few minutes. However, the best antimicrobial action was shown by the cotton fabric coated with Fe_2_O_3_ at 200 W for 120 min. The current research aimed to evaluate the antimicrobial effect of the cotton fabric coated with Fe_2_O_3_ on both sides against 16 microbial strains (Gram-positive and Gram-negative bacteria and fungi).

## 2. Materials and Methods

### 2.1. Preparation of Samples

Cotton fabric (100% cotton, density—138 gsm) was used as the test textile material. An unmodified cotton textile control was autoclaved before use. The material was cut into squares of 10 mm × 10 mm using sterile scissors and then placed in a sample holder. All cotton fiber samples were tested for sterility using a thioglycolate medium (CM0173, Thermo Fisher Scientific, Basingstoke, UK) before the deposition of NPs in low-temperature plasma.

During Fe_2_O_3_ NP deposition, the cotton fabric was placed in the middle of two iron electrodes (produced by KJLC, Clairton, PA, USA) with 99.9% purity. The stainless-steel frame was made from wires and used as a cotton fabric holder. The scheme of the deposition procedure is presented in Figure 1. Pulsed DC (from 20 W to 200 W) was applied for low-temperature plasma generation and pure oxygen (99.999%) gas was used as the working gas. The working pressure during the deposition was 10 Pa and kept stable all the time. The distance between the iron electrodes and cotton fabric substrates was 5 cm and the deposition time was from 5 to 120 min.

NP deposition was also performed on glass slide substrates in the same chamber as the separate deposition experiments. Glass slide substrates were placed instead of cotton fabric substrates and deposition of NPs was performed using the same procedures as for textile samples.

### 2.2. Characterization of Materials

Surface morphology analysis was performed using a scanning electron microscope (SEM, Hitachi S-3400 N, Tokyo, Japan) equipped with a secondary electron detector. The elemental composition and the elemental mapping of the substrates coated with Fe_2_O_3_ nanoparticles were investigated employing energy-dispersive X-ray spectroscopy (EDS, Bruker Quad 5040, Billerica, MA, USA). XPS measurements were performed using a PHI Versaprobe spectrometer (Al monochromator, 25 W beam power, 100 µm beam size). The energy scale of the instrument was calibrated using Au 4f7/2 (84.0 eV) and Cu 2p3/2 (932.6 eV) peaks. A dual charge neutralization system consisting of low-energy electron and ion sources was used to compensate sample charging. The prevailing charge effect was compensated by aligning the adventitious carbon C1 component to 284.8 eV. Atomic force microscopy (AFM, NT-206, Microtestmachines, Gomel, Belarus) measurements were performed for the surface topography investigations. The main AFM parameters were maximum range of heights: 2–4 μm; lateral resolution (plane XY): 1–5 nm; vertical resolution (direction Z): 0.1–0.5 nm; scanning matrix: up to 1024 × 1024 points.

### 2.3. Collection of Microbial Strains

A total of 16 reference, clinical, and zoonotic strains of Gram-negative and Gram-positive bacteria and yeasts were used. Zoonotic strains *E. cloaceae*, *S. enterica*, *S. haemolyticus*, and *Corynebacterium* spp. Were isolated from diseased animals, whereas *S. hominis*, *S. epidermidis*, *C. freundii*, *A. baumannii*, *E. faecium*, and *C. acnes* were isolated from diseased or healthy humans.

Susceptibility testing of microorganisms, previously isolated at the Microbiology and Virology Institute of the Lithuanian University of Health Sciences, according to the EUCAST guidelines [21], was performed. Strains had the following resistance to antibiotics: *Enterobacter cloaceae* (*E. cloacae*) (resistance: ampicillin, sulfamethoxazole/trimethoprim, gentamicin, cefoxitin), *Salmonella enterica* (*S. enterica*) (resistance: none), *Citrobacter freundii* (*C. freundii*) (resistance: ampicillin, ciprofloxacin, cefoxitin, amoxicillin/clavulanic acid), *Acinetobacter baumannii* (*A. baumannii*) (resistance: gentamicin, ciprofloxacin, amikacin, imipenem, meropenem), *Staphylococcus haemolyticus* (*S. haemolyticus*) (resistance: penicillin, erythromycin, cefoxitin, ciprofloxacin), *Staphylococcus hominis* (*S. hominis*) (resistance: benzylpenicillin, trimethoprim-sulfamethoxazole), *Staphylococcus epidermidis* (*S. epidermidis)* (resistance: benzylpenicillin, erythromycin, trimethoprim-sulfamethoxazole, cirpfloxacin), *Corynebacterium* spp. (resistance: benzylpenicillin, clindamycin), *Cutibacterium acnes* (*C. acnes*) (resistance: benzylpenicillin, vancomycin, clindamycin, ampicillin, imipenem), and *Enterococcus faecium* (*E. faecium*) (resistance: penicillin, tetracycline, quinupristin/dalfopristin). ATCC strains tested included *Klebsiella pneumoniae* (*K. pneumoniae*) ATCC 10031, *Pseudomonas aeruginosa* (*P. aeruginosa*) ATCC 27853, *Aeromonas hydrophila* (*A. hydrophila*) DSM 112649, *Staphylococcus aureus* (*S. aureus*) ATCC 25923, *Enterococcus faecalis* (*E. faecalis*) ATCC 29212, and *Candida albicans* (*C. albicans*) ATCC 10231.

### 2.4. Determination of Antibacterial Activity of Cotton Fabric Textile Coated with Fe_2_O_3_

The antibacterial activity of the cotton fabric with two sides coated with Fe_2_O_3_ 200 W was checked against reference strains *S. aureus* ATCC 25923 and *K. pneumoniae* ATCC 10031.

A 0.5 McFarland turbidity standard of *S. aureus* and *K. pneumoniae* was prepared using a physiological solution. Squares of treated fabrics with different metal nanoparticles (10 mm × 10 mm) were placed into an empty sterile Petri dish, and then 50 µL of bacterial suspension was loaded onto the treated fabric. Plates with loaded squares were covered and kept at room temperature and under laboratory lighting conditions for 60 min according to previously published data. One hour is sufficient time for inactivation of most microorganisms using coated material with metal oxide nanoparticles [7].

Additionally, a physiological solution was prepared for 10-fold dilutions of bacterial culture—the first tube with 10 mL of physiological solution, the rest of them with 9 mL of physiological solution. Squares were kept for an hour in Petri dishes, then were transferred into a tube with 10 mL of physiological solution. The tube was vortexed, and then 10 mL of liquid was transferred to a tube with 9 mL of physiological solution. The same action was repeated three times. The 10-fold dilutions were from 10^1^ to 10^−4^. A total of 50 µL of liquid from ten-fold dilutions of the samples were plated on soya agar (three plates for each dilution). Plates were incubated in a thermostat for 24 h at a temperature of 36 °C for bacteria and 48 h of 30 °C for yeasts. Colony counts of *S. aureus* and *K. pneumoniae*, and later for all 16 strains, were performed. The average number of counted colonies was calculated from three plates. Sterile squares of sterile cotton fabric were used as a control.

### 2.5. Statistical Analysis

Statistical analysis was performed using the R statistical package, version 3.6.2 [22]. Results were considered statistically significant when *p* < 0.05. Graphics were performed using the SPSS/W 29.0 software (Statistical Package for the Social Sciences for Windows, Inc., Chicago, IL, USA).

## 3. Results and Discussion

### 3.1. Deposition of Fe_2_O_3_ Nanoparticles on the Cotton Fabric

Functional Fe_2_O_3_ coatings were deposited on the two sides of the fabric. Figure 2 presents SEM morphology views at different magnifications.

Surface morphology analysis revealed that the initial textile consists of uniform fibers (Figure 2a,c), which become covered with uniform structures after immersion in low-temperature plasma (Figure 2b,d).

Elemental mapping was performed to view the distribution of the elements on the textile surface. It shows (Figure 3) that the Fe and O distribution is uniform, and it confirms the presumption that iron oxide does not form a continuous film on the textile fiber surface but stays as separate homogeneously distributed clusters.

Similar results were reported by the other authors confirming that iron oxide distributed homogeneously on the textile surface in the form of nanoparticles using the pad-dry-cure process [23].

### 3.2. Deposition of Fe_2_O_3_ Nanoparticles on the Glass Slide

To further understand the geometry and distribution of Fe_2_O_3_ NPs on the surface, deposition was performed on glass slide substrates. All NP synthesis parameters were kept similar to the synthesis conditions where Fe_2_O_3_ was deposited on the textile surface.

The EDS results are presented in Table 1. The analysis performed confirmed the presence of Fe particles (up to 1.96 at. %) on the surface after deposition. The concentration of all other materials, such as Na, Mg, Ca, etc., changed very little and that confirms our presumption that Fe_2_O_3_ forms a very thin, non-uniform, altered layer on the glass slide surface.

Elemental mapping was performed to see the distribution of the elements on the glass slide surface. It shows (Figure 4) that the Fe distribution is uniform, but due to low density, it could be presumed that Fe does not form uniform film but stays as separate islands on the glass slide surface.

The surface topographies were analyzed using AFM before and after Fe_2_O_3_ NP deposition and are shown in Figure 5.

Surface topography analysis before and after deposition revealed that the initial substrate surface (Figure 5a) is rather smooth with the presence of individual irregularities, which highly influenced the roughness parameter (R_a_ = 0.836 nm). After plasma treatment and formation of Fe_2_O_3_ nanostructures on the surface (Figure 5b), it becomes covered with spike-like structures. Surface roughness increases up to 50% and the R_a_ value becomes 1.242 nm. This could be caused by the fact that iron oxide does not form uniform films on the surface; instead, it forms cluster-like structures. Cluster-like structure formation could be explained due to a low deposition temperature during the deposition process (as textile samples can withstand it without any essential aging effects) and minimal impact of ion bombardment from plasma as the sample holder does not have any applied potential. As a result, the migration of arrived atoms and atom groups from plasma remains minimal on the surface, and instead of uniform films, the formation of iron oxide spike clusters on the surface was observed.

Elemental composition and formation of compounds were further investigated using the XPS analysis technique. Figure 6 shows XPS measurements of the initial glass slide surface before deposition. This survey scan is very important, confirming that there are no Fe_2_O_3_ NPs on the glass slide surface before the deposition process.

Fe, O, C, and Na elements were detected on the deposited non-treated sample surface (Figure 7). An additional experiment was performed to determine surface elemental composition after the removal of possible contamination, which could occur after sample extraction from the deposition chamber and transfer to the XPS chamber. It was carried out using surface sputtering by an Ar^+^ ion gun. Firstly, more specifically, four relatively gentle sample sputtering steps were applied (duration 1 min, acceleration voltage 2 kV, raster size 2 mm× 2 mm). After each sputtering step, a new survey spectrum was acquired. The same four elements were observed in all the cases. Lastly, we increased the ion gun acceleration voltage to 4 kV and performed one more sputtering for 1 min with the same ion beam rastering. By comparison, XPS survey spectra were performed before and after all ion gun sputtering steps (Figure 7 (blue curve)); it was noticed that the intensity of Fe and O peaks increased, whereas the intensity of C1 peaks were relatively like the initial ones. The increase of Fe and O peak intensity is natural as surface cleaning from adventitious contamination reveals a “true” Fe–O film surface. The fact that intensities of carbon and sodium peaks are not diminishing indicates that uniform Fe–O does not form on the surface, and it more probably stays in cluster-like structures.

XPS analysis confirmed the presence of Fe on the top layer (Table 2). Thus, combining AFM and XPS observations, it could be concluded that deposited Fe_2_O_3_ does not form uniform thin films on the surface.

High-resolution Fe 2p, O 1s, and C 1s core electron spectra from not sputtered Fe–O samples are provided in Figure 8. Notably, the key Fe 2p3/2 peak component energy of 710.9 eV as well as O 1s component peak energy of 530.2 eV provide a good fit with the commonly reported corresponding energies for Fe_2_O_3_ oxide.

C 1s spectrum has components at 283.7 eV, 284.8 eV, 286.0 eV, and 289.2 eV. These peaks can be attributed to Fe–C, C–C, C–O, and C=O bonds, respectively. The presence of a high fraction of C–O and C=O bonds shows that some carbon is incorporated into the iron oxide structure during deposition. It comes from the working gas atmosphere and, due to C affinity to Fe, is trapped inside the growing Fe_2_O_3_ structure.

### 3.3. Antimicrobial Activity of Cotton Fabrics with Two Sides Coated with Fe_2_O_3_ NPs against 16 Microorganisms

Firstly, the antimicrobial properties of cotton fabric with one and two sides coated with Fe_2_O_3_ nanoparticles were tested against two strains—*K. pneumoniae* and *S. aureus.* Two-side coated textiles had high antibacterial activity against both Gram-negative (*K. pneumoniae*) and Gram-positive (*S. aureus*) bacteria. The inhibition of bacterial growth was more than 80% and 90%, respectively. Thus, two-side coated Fe_2_O_3_ cotton textiles compared with one-side coated textiles had a higher inhibitory effect on both strains of bacteria (respectively, *p* = 4.989 × 10^−7^ and *p* < 2.2 × 10^−16^). The one-side coated textile inhibited the growth of *K. pneumoniae* but it promoted the growth of *S. aureus.* A possible explanation for *S. aureus* growth activation might be due to the additional amount of iron on the one-side coated textile. It is known that iron metal starvation represses the growth of aerobically respiring *S. aureus* [24]. However, elevated levels of iron are toxic to cells because of their ability to promote the formation of damaging oxidative radicals [25], which can lead to bacterial death.

The differences in antimicrobial effect on *S. aureus* ATCC 25923 and *K. pneumoniae* ATCC 10031 between one-side and two-side coated samples are presented in Figure 9.

Secondly, evaluation of the antimicrobial activities of a cotton fabric on both sides coated with Fe_2_O_3_ 200 W was performed with 16 reference and clinical/zoonotic strains—bacteria (15 strains) and fungus (1 strain). It was determined that Fe_2_O_3_ 200 W had good antibacterial action on both Gram-positive and Gram-negative bacteria. Opposite to our previous study with CuO nanoparticles [7], Fe_2_O_3_ nanoparticles coated on two sides of cotton fabric effectively inhibited the growth of fungi *C. albicans* and reduced the growth by 95%. The results of the research are presented in Figure 10.

Figure 11 demonstrates SEM micrograph images after bacteria *K. pneumoniae* was transferred onto a glass slide without Fe_2_O_3_ NPs (control) and Fe_2_O_3_ NPs coated glass slide. It could be seen from the picture that cells, after contact with Fe_2_O_3_ NPs, have membrane damage. As reported by Slavin et al., 2017, the exposure of NPs to bacterial cells can lead to membrane damage caused by NP adsorption, sometimes followed by penetration into the cell [26].

The current study on the antimicrobial activity of cotton fabric coated with Fe_2_O_3_ nanoparticles revealed that the effect depends on the deposition parameters, such as power source, deposition duration, and one- or two-sided coated material. Fe_2_O_3_ (*p* = 200 W) nanoparticles coated on two sides of cotton fabric had antibacterial activity on *K. pneumoniae* and *S. aureus*. The effectiveness of Fe_2_O_3_ nanoparticle’s antibacterial effect against *S. aureus* was reported in the recent publication of Alangari et al., 2022, additionally with good activity towards the tested fungal strain *Candida albicans* [27]. Cotton fabric treated on two sides with Fe_2_O_3_ 200 W nanoparticles showed better antimicrobial properties compared to single-side coated samples with metal oxide nanoparticles. The nanoparticles Fe_2_O_3_ 200 W coated on both sides of the cotton textile stopped the growth of all bacteria *K. pneumoniae* and *S. aureus.* While single-sided cotton material was less effective against Gram-positive bacteria, it showed a good antibacterial effect against Gram-negative bacteria. This might depend on the deposition of different amounts of Fe_2_O_3_ 200 W onto the fabric surface.

## 4. Conclusions

In this work, pure Fe_2_O_3_ nanoparticles were successfully deposited on textile and glass slide substrates using low-temperature plasma technologies. Surface morphology and topography analysis confirmed that nanoparticles formed cluster-like structures and distributed homogeneously on the substrate’s surfaces.

The obtained results demonstrated that textiles coated with Fe_2_O_3_ NPs showed a high antimicrobial effect on Gram-negative and Gram-positive bacteria as well as on fungi *C. albicans*. In total, 100 percent of inactivation was towards staphylococci (including MRSA), *E. faecalis*, *A. baumanii*, *S. haemolyticus*, *K. pneumoniae*, and *A. hydrophila*. Coated textiles with Fe_2_O_3_ NPs were also effective against yeasts (95.4% of inactivation).

These results show great potential for the use of low-temperature plasma deposition technologies for antimicrobial nanotextile production and widespread use in various products, facilities, and settings.

## Figures and Tables

**Figure 1 nanomaterials-13-03106-f001:**
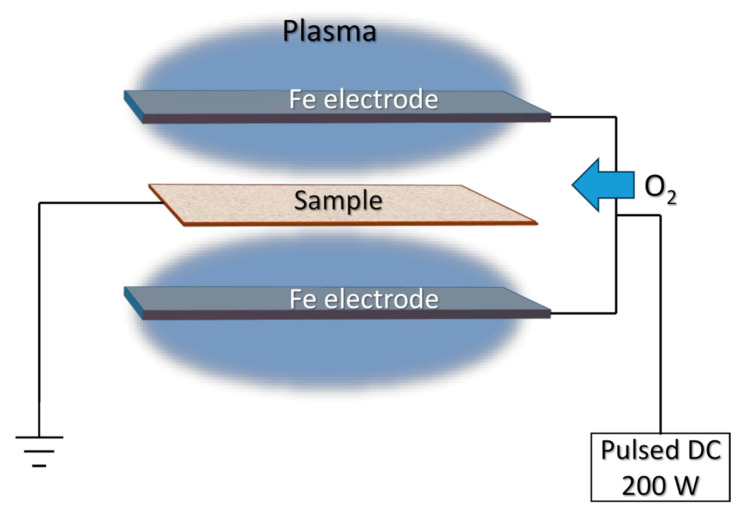
Deposition setup for Fe_2_O_3_ NP deposition on cotton fiber and glass slide substrates.

**Figure 2 nanomaterials-13-03106-f002:**
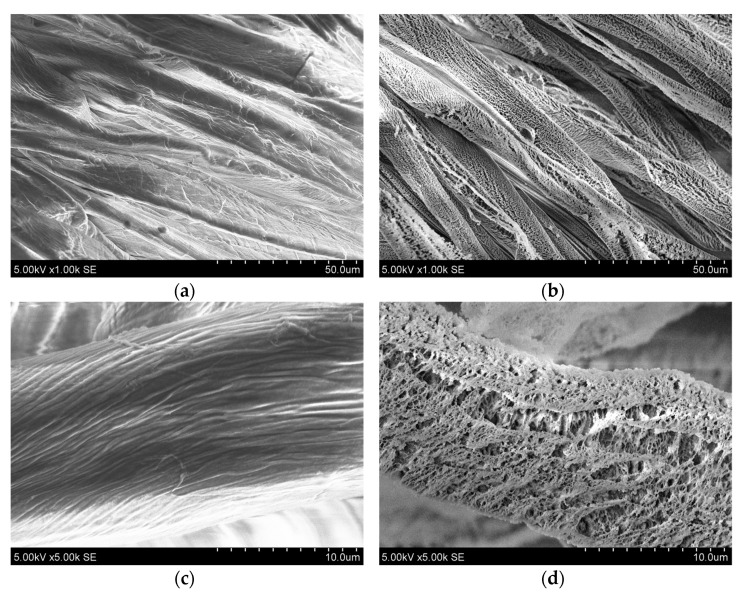
Cotton fabric surface morphology views of initial (**a**,**c**) and Fe_2_O_3_ coated textile surfaces (**b**,**d**) at different magnifications (1.00k and 5.00k, respectively).

**Figure 3 nanomaterials-13-03106-f003:**
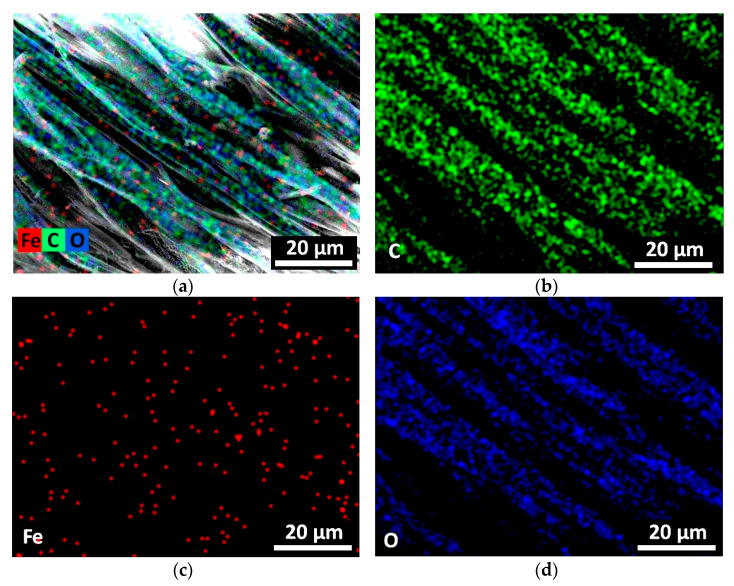
The elemental mapping view of all C, Fe, and O elements together (**a**) and deposited separately on textile surface: element C (**b**), element Fe (**c**), and element O (**d**).

**Figure 4 nanomaterials-13-03106-f004:**
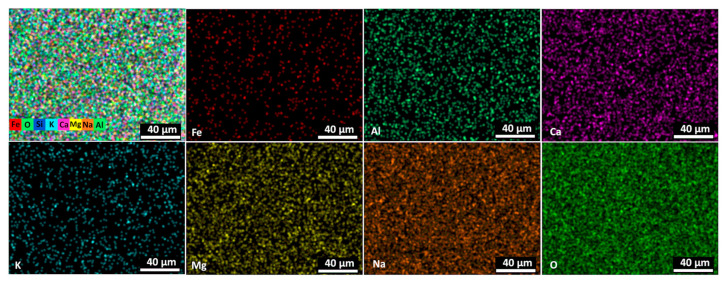
The elemental mapping view of Fe, Al, Ca, K, Mg, O and Na as all elements and separately on glass slides—Fe, Al, Ca, K, Mg, Na.

**Figure 5 nanomaterials-13-03106-f005:**
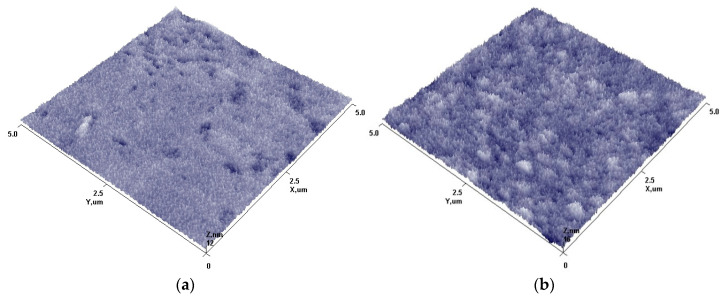
The surface topography before and after deposition of Fe_2_O_3_ NPs on the glass slides. Initial glass slides surface (**a**), glass slides covered with Fe_2_O_3_ nanoparticles (**b**).

**Figure 6 nanomaterials-13-03106-f006:**
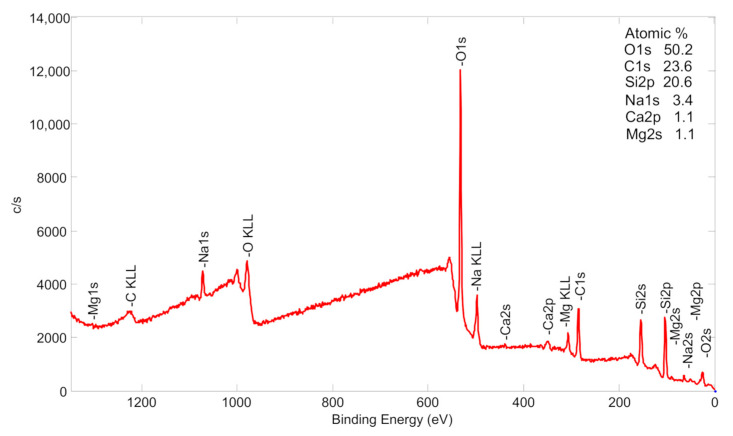
A survey spectrum of glass slide samples before Fe_2_O_3_ deposition is presented (red curve).

**Figure 7 nanomaterials-13-03106-f007:**
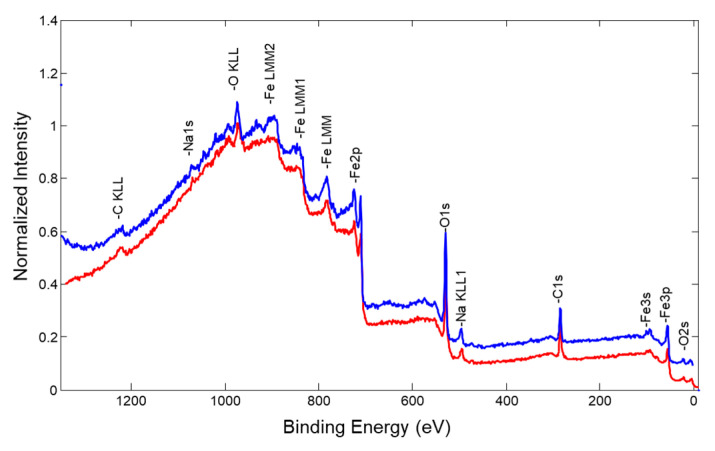
XPS survey spectra of Fe–O sample: red—initial surface, blue—sample surface after Ar^+^ ion gun sputtering.

**Figure 8 nanomaterials-13-03106-f008:**
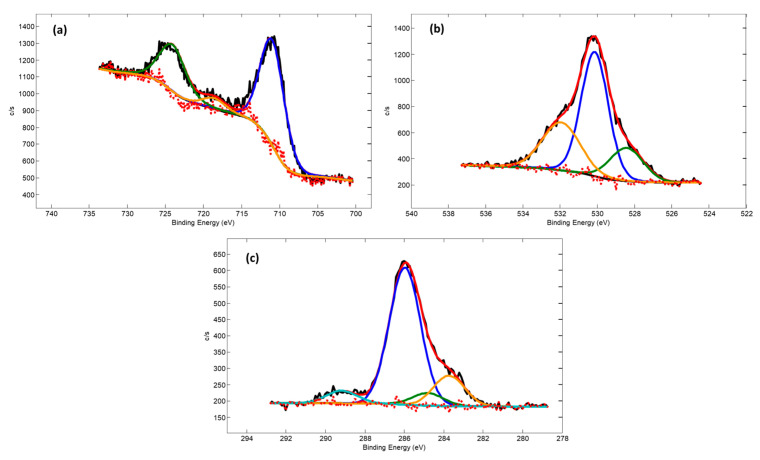
Core level photoelectron spectra: Fe 2p (**a**), O 1s (**b**), and C 1s (**c**).

**Figure 9 nanomaterials-13-03106-f009:**
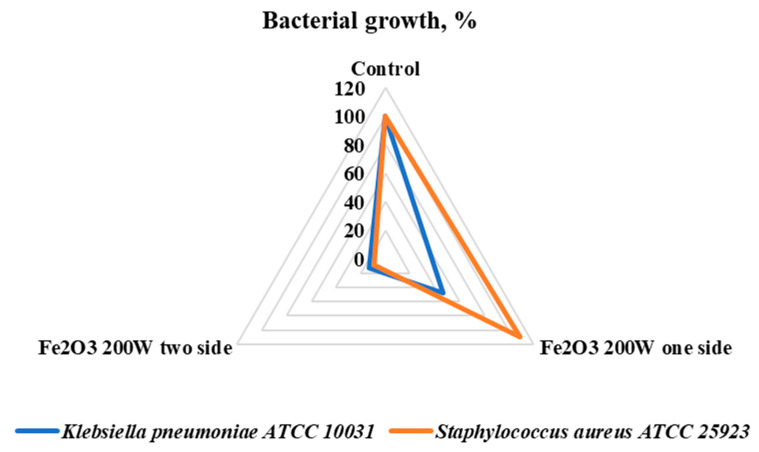
Antimicrobial effect of cotton fabric coated on one side and two sides with Fe_2_O_3_ nanoparticles on *S. aureus* ATCC 25923 and *K. pneumoniae* ATCC 10031.

**Figure 10 nanomaterials-13-03106-f010:**
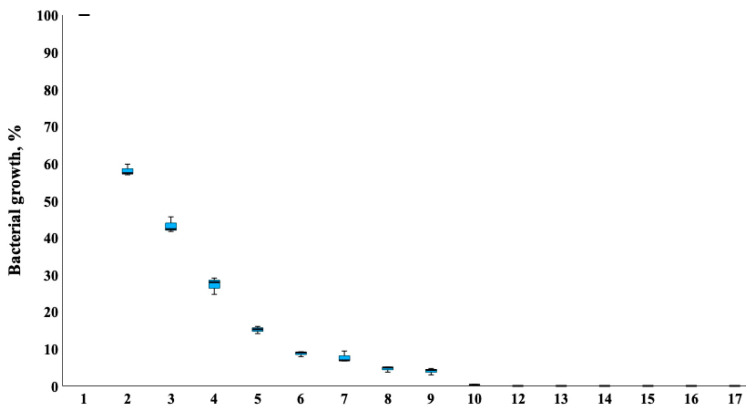
Antimicrobial effect of cotton fabric with two sides treated with Fe_2_O_3_ oxides nanoparticles on 16 strains of microorganisms: 1—control, 2—*S. enterica*, 3—*Corynebacterium* spp., 4—*P. aeruginosa ATCC 27853*, 5—*E. cloacae*, 6—*C. acnes*, 7—*S. hominis*, 8—*C. albicans* ATCC 10231, 9—*E. faecium*, 10—*S. epidermidis*, 11—*C. freundii*, 12—*S. aureus* ATCC 25923, 13—*E. faecalis* ATCC 29212, 14—*A. baumannii*, 15—*S. haemolyticus*, 16—*K. pneumoniae* ATCC 10031, 17—*A. hydrophila* DSM 112649.

**Figure 11 nanomaterials-13-03106-f011:**
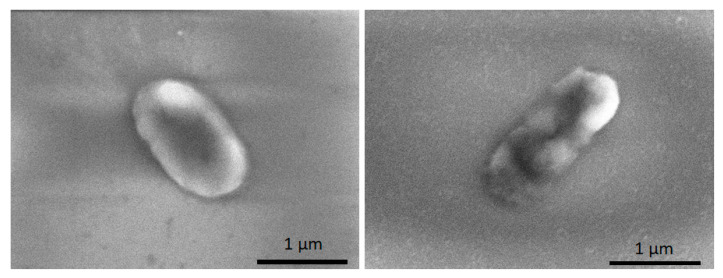
*K. pneumoniae* cells: unaffected (**left**) and damaged by Fe_2_O_3_ NPs (**right**).

**Table 1 nanomaterials-13-03106-t001:** EDS elemental concentration of samples’ top layer.

Sample	Top Layer Elemental Composition, at. %								
	Si	O	Fe	Na	Mg	Ca	K	S	Al
Before deposition	24.06	60.19	0	9.66	2.24	2.69	0.50	0.14	0.52
After Fe_2_O_3_ NP deposition	26.23	56.06	1.96	9.44	2.39	2.60	0.65	0	0.68

**Table 2 nanomaterials-13-03106-t002:** XPS elemental concentration of samples’ Fe–O top layer.

Sample	Top Layer Elemental Composition, at. %						
	O	C	Fe	Si	Na	Ca	Al
Before deposition	50.2	23.6	-	20.6	3.4	1.1	1.1
After Fe_2_O_3_ NP deposition	41.1	40.0	17.9	-	1.0	-	-

## Data Availability

The data presented in this study are available on request from the corresponding author. The data are not publicly available, due to the next work.

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
