# Peer review of "Development of Antibacterial Cotton Textiles by Deposition of Fe2O3 Nanoparticles Using Low-Temperature Plasma Sputtering"

_nanomaterials, 2023, doi:10.3390/nano13243106_

Round 1

Reviewer 1 Report

Comments and Suggestions for Authors

This work is relevant and interesting for the field. However, there are some issues that should be taken into account:

1. In the introduction section the sentence "Fe2O3 nanoparticles can also be produced using wet chemical methods, and combined with other materials, such as Ag, to enhance their antibacterial and antifungal activities [19]."  should be integrated in the previous paragraph, as it is not related to the objective of the work described afterward. 

2. Figures 9 and 11 lack error bars or standard deviation. Information on this aspects and on the number of independent assays conducted should be added to improve the reliability of the data.

3. What is the reason for the single cotton fabric being less effective against gram positive bacteria and the opposite occurring for cotton fabric coated on both sides ? Additional discussion on this topic is necessary. 

Comments on the Quality of English Language

I believe the manuscript needs major revisions (or presentation of data that was not clearly presented on the main manucript) before acceptance particularly on the assays performed with bacteria which lack error bars. 

Reviewer 2 Report

Comments and Suggestions for Authors

In this work, the nanoparticles were deposited using low-temperature plasma technology in a pure oxygen atmosphere to enhance the cotton fabric’s antimicrobial properties by depositing Fe2O3 nanoparticles on both sides of its surface. The obtained results demonstrated that textiles coated with Fe2O3 NPs showed a certain antimicrobial effect on some kinds of bacteria and fungi. However, there are still some issues that need to be explained.

Q1: What are Figures 9 and 10 intended to showcase? Please provide a detailed explanation. The two figures seem to show the same content and it is suggested that the illustrations could be combined.

Q2: What’s the meaning if some carbon is incorporated into the iron oxide structure, in figure 8? Please provide data explanation.

Q3: In Figure 11, the Author described:” It was determined that Fe2O3 200W works better on gram-positive bacteria than on gram-negative (p=8.034×10-13). Fungi Candida albicans (C. albicans) was inhibited with cotton fabric coated on two sides with Fe2O3 nanoparticles.” However, the bacteria and fungi mentioned above, cannot find in Figure 11, please explained.

Q4: The whole manuscript needs to be polished and some English grammar to be fixed.

Comments on the Quality of English Language

it is okay

Reviewer 3 Report

Comments and Suggestions for Authors

Authors deposited Fe2O3 nanoparticles on the cotton fabric by using a low temperature plasma technology to improve its antimicrobial activity. The conditions for the nanoparticle deposition were optimized; the antimicrobial activity of cotton textiles against different bacteria and fungi were detected. It is very interesting and important work. Some questions need to resolve before publishing:

Line 18: antibacterial should be antimicrobial due to its activity against bacteria and fungi.

In 2.4 section: Please give some references of incubation for 1 h in Line 148-149 if it is possible. Plates were incubated in a thermostat for 24 h at a temperature of 36ºC, in which the incubation time is commonly 16-18 h for bacteria, but for fungi, it should be lower temperature and longer time.

Please give more information of bacteria, which are clinical strains, from hospital or animal farms, etc. if it is possible?

What about the price of Fe2O3 materials?

In SEM images, the bacterial membrane damaged by Fe2O3 NPs is not significant, could you please provide more bacterial cells in same image? And provide higher quality of images if it is possible.

Others:

Line 165, 222, 260: Fe2O3

Line 260-261, 262: italic S. aureus and K. pneumoniae.

Line 269: fungi should be fungus.

Round 2

Reviewer 1 Report

Comments and Suggestions for Authors

I believe the assays performed with bacteria lack significance. More than one independent assays should be conducted. Error bars look all the same for every condition tested in Figure 10, which does not reflect the variability of this type of assays. 

Comments on the Quality of English Language

The quality of english was improved

Round 3

Reviewer 1 Report

Comments and Suggestions for Authors

All my comments were addressed so I believe the manusccript can be accepted.